# Pre- and Postoperative Exercise Effectiveness in Mobility, Hemostatic Balance, and Prognostic Biomarkers in Hip Fracture Patients: A Study Protocol for a Randomized Controlled Trial

**DOI:** 10.3390/biomedicines11051263

**Published:** 2023-04-24

**Authors:** Apostolos Z. Skouras, Dimitrios Antonakis-Karamintzas, Charilaos Tsolakis, Argirios E. Tsantes, Georgia Kourlaba, Ioannis Zafeiris, Fotini Soucacos, Georgios Papagiannis, Athanasios Triantafyllou, Dimitra Houhoula, Olga Savvidou, Panagiotis Koulouvaris

**Affiliations:** 11st Department of Orthopaedic Surgery, School of Medicine, National and Kapodistrian University of Athens, 12462 Athens, Greece; apostolis.sk@gmail.com (A.Z.S.);; 2Sports Performance Laboratory, School of Physical Education & Sports Science, National and Kapodistrian University of Athens, 17237 Athens, Greece; 3Laboratory of Haematology and Blood Bank Unit, “Attiko” Hospital, School of Medicine, National and Kapodistrian University of Athens, 12462 Athens, Greece; 4Faculty of Health, Department of Nursing, University of Peloponnese, 23100 Sparta, Greece; 5Biomechanics Laboratory, Department of Physiotherapy, University of the Peloponnese, 23100 Sparta, Greece; 6Department of Food Science and Technology, University of West Attica, 12244 Egaleo, Greece

**Keywords:** hip fracture, exercise therapy, prehabilitation, rehabilitation, physiotherapy, coagulation, fibrinolysis, rotational thromboelastometry

## Abstract

Hip fractures are a major health concern, particularly for older adults, as they can reduce life quality, mobility loss, and even death. Current evidence reveals that early intervention is recommended for endurance in patients with hip fractures. To our knowledge, preoperative exercise intervention in patients with hip fractures remains poorly researched, and no study has yet applied aerobic exercise preoperatively. This study aims to investigate the short-term benefits of a supervised preoperative aerobic moderate-intensity interval training (MIIT) program and the added effect of an 8-week postoperative MIIT aerobic exercise program with a portable upper extremity cycle ergometer. The work-to-recovery ratio will be 1-to-1, consisting of 120 s for each bout and four and eight rounds for the pre- and postoperative programs, respectively. The preoperative program will be delivered twice a day. A parallel group, single-blinded, randomized controlled trial (RCT) was planned to be conducted with 58 patients each in the intervention and control groups. This study has two primary purposes. First, to study the effect of a preoperative aerobic exercise program with a portable upper extremity cycle ergometer on immediate postoperative mobility. Second, to investigate the additional effect of an 8-week postoperative aerobic exercise program with a portable upper extremity cycle ergometer on the walking distance at eight weeks after surgery. This study also has several secondary objectives, such as ameliorating surgical and keeping hemostatic balance throughout exercise. This study may expand our knowledge of preoperative exercise effectiveness in hip fracture patients and enhance the current literature about early intervention benefits.

## 1. Introduction

A hip fracture is the leading cause of morbidity in people 65 and older, ranking it in the top ten causes of person/year-adjusted disability for older adults [1]. In Greece, the majority of hospitalized patients with a hip fracture are over 75 years old [2]. Globally, the number of people with hip fractures is expected to increase significantly due to the aging population, and other factors, such as osteoporosis, partly attribute to insufficient physical activity [1]. In the US, according to data from the Centers for Disease Control and Prevention (CDC), one in three who practiced self-care before sustaining a hip fracture had to be institutionalized in a nursing home for at least one year after the fracture [3]. In Greece, between 1977 and 2007, hip fractures doubled, with the annual incidence rising to 343.96 per 100,000 inhabitants [4]. For the year 2019, Greece, in terms of the percentage of the population over 80, was in second place among the countries of Europe, with this age group constituting 7.1% of its total population. A prospective Greek cohort study documented an increased relative risk by 9% and 10% of 5- and 10-year mortality for each added year in age [5]. Evidence shows that patients with a hip fracture have a significantly higher risk of death up to one year after the fracture than those without a hip fracture. Up to 15 times higher mortality within one month after a hip fracture compared to non-fractured peers has been reported in the literature, not attributable to poor pre-fracture health status [1].

Many factors and biomarkers have been linked to survival and functional outcomes postoperatively in patients with hip fractures. For example, an elevated blood lactate concentration has been implicated in increased immediate or mid-term postoperative mortality during surgery [6]. Similarly, decreased albumin levels are a strong independent predictor of mortality after a hip fracture, and prealbumin also appears to be an important marker due to its better sensitivity to changes [7]. Examining in a Bayesian Belief Network structure the predictors’ survival 1 year after hip fracture surgery, the number of comorbidities on admission (≥2 conditions) and levels of nitrogen (≥9 mmol/L), albumin (<35 g/L) and lactate (≥2 mmol/L) in the blood, are the four first-rate predictors of survival [8]. Moreover, venous thrombosis remains one of the main reasons for readmission to the hospital in the first 30 days after surgery [9]. Blood coagulation involves complex pathways which regulation depends on several factors. There is clear evidence that hip fractures are associated with blood hypercoagulability, and pharmaceutical and mechanical prophylaxis is necessary [10,11,12,13]. Finally, delayed surgery has been associated in many observational studies with increased mortality and serious complications [14].

People recovering from a hip fracture are also at a high risk of muscle weakness and fear of falling and experience limitations in mobility, self-care, and participation in the community [1]. These consequences last months after surgery, with 50% experiencing reduced functionality within one year, while only 30% regain full functionality [15]. Exercise has significant benefits for several of these variables, such as the level of independence and self-care [16,17], the faster clearance of accumulated lactate [18,19], the increased activity of pyruvate dehydrogenase [20] and the immediate enhancement of cognitive functions [21]. Hemostasis is also linked to exercise physiology, as it affects blood coagulation, fibrinolysis and platelets [22]. It has been shown that exercise promotes fibrinolysis, with increased levels of tissue plasminogen activator (t-PA), decreased fibrinogen and decreased plasminogen activator inhibitor (PAI-1). Furthermore, following specific exercise protocols has been shown to decrease surrogate markers of thrombin and endogenous thrombin potential (ETP) [23,24,25]. Regular aerobic exercise is associated with a decreased resting platelet aggregation and activation and thrombomodulin for platelet function, and there is an indication of a reduced thrombin–antithrombin complex [26]. The observed exercise-induced changes in the coagulation factors occur independently of the older adult’s lipid profile, aerobic capacity and body composition [27].

In a recent umbrella review, early intervention seems to be beneficial for improving the endurance in hip fracture patients [28]. However, it is not yet clear if early intervention has any effect on functioning, quality of life, length of stay and mortality in this specific population. In order to prevent mortality and morbidity, preoperative physical therapy programs have been implemented in various conditions, such as in cardiac operations [29], in non-small cell lung cancer [30] and in major abdominal surgery [31], as well as in various other orthopedic surgeries (e.g., knee and hip arthroplasty) [32]. To the best of our knowledge, preoperative physical therapy intervention in patients with hip fractures remains poorly researched, and no study was found in the literature with the application of aerobic exercise preoperatively. The study aims to investigate the short-term benefits of a preoperative aerobic exercise program and the added effect of an 8-week postoperative aerobic exercise program with a portable upper extremity cycle ergometer.

## 2. Materials and Methods

### 2.1. Study Design

The study protocol has been prospectively registered in the open database for study protocols, ClinialTrials.gov (registration number: NCT05389800) [33]. The study received approval number 402/26-7-2021 from the Institutional Review Board of the “Attikon” University Hospital (1st Department of Orthopaedic Surgery, National and Kapodistrian University of Athens) 7 September 2021.

A parallel group, single-blinded, randomized controlled trial (RCT) will be conducted with two groups of hip fracture patients. The sample will be recruited from a tertiary University Hospital (1st Department of Orthopaedic Surgery, “Attikon” University Hospital, Athens, Greece). The randomization will be done at the time of admission of the patients to the emergency department (ED) and after their written informed consent form. The study protocol’s instructions will be described through an informed consent form in plain language, along with all the procedures the subjects will undergo and the risks and benefits of participation. Allocation concealment will be ensured by using sequentially numbered, opaque, sealed envelopes. The envelopes were created by a biostatistician, who will participate actively in neither the delivery of any intervention nor in the outcome measures assessment. For the preoperative program, 116 patients need to be recruited, who will be randomized with a ratio of 1:1 using the block randomization method (58 in the intervention group and 58 in the control group). Immediately after the surgery, the participants of each group will continue in the postoperative intervention program: the intervention group (IG) will continue with the exercise program three times a week for a total of 8 weeks, while the control group (CG) will not receive any intervention at any time point after surgery (Figure 1, graphic illustration).

### 2.2. Implementation of the Study

#### 2.2.1. Getting Informed Consent

Participants will be informed of the purpose of the study, and their signed consent will be requested at the start of the study after it has been ensured that they meet the inclusion criteria and do not meet any of the exclusion criteria. Participation in the study is voluntary, and participants may discontinue participation at any time.

#### 2.2.2. Regulatory Study Approval

The present study is an interventional study that is not subject to government decision D3(a)—36809 (Government Gazette B 2015, 3-6-2019); therefore, it has not been submitted to the National Medicines Agency (ΕOΤ), and an approval from the ethics committee is sufficient and ethics of the center/body conducting the study. The present study has been designed following the rules of research ethics and ethics resulting from the Declaration of Helsinki.

#### 2.2.3. Final Report and Publications

A final report with study results will be prepared after this research study. Publications will comply with the International Committee of Medical Journal Editors (ICMJE) guidelines. Negative and positive results should be published or made available publicly. Regarding authors, the following criteria should be met:Authors must make substantial contributions to study the conception and design, data acquisition or data analysis and interpretation of the results.Authors must draft the article for publication or, during authorship review, contribute (data analysis, interpretation or other important intellectual content) to the extent that leads to a significant revision of the article in agreement with the other authors.Authors must provide written approval of the article’s final version before submission.

### 2.3. Participants

The study will include both male and female participants, making the research more representative and diverse. All participants will be 65 years and older, highlighting a focus on the geriatric population who more regularly suffer hip fractures due to falls. The study will specifically target individuals undergoing surgery for hip fractures, providing valuable insights into the management and outcomes of this patient population. A total of 116 patients will be enrolled in the study, with 58 patients assigned to each group, namely the intervention and control groups. This equal allocation ensures a balanced comparison of the outcomes between the two groups.

### 2.4. Eligibility Criteria

Of the patients hospitalized at the 1st Department of Orthopaedic Surgery in the “Attikon” University Hospital (Athens, Greece), candidates to participate in the study will meet all of the following inclusion criteria and not meet any of the exclusion criteria. To mitigate attrition, the exercise program will be individualized, and there will be regular follow-up assessments. The postoperative exercise program will also be delivered at home; thus, it will be easier for patients to engage in it as planned. Patients will not receive any fee for participating in the study.

#### 2.4.1. Inclusion Criteria

Age: 65 years old and olderUnilateral proximal femoral fracture/hip fracture (intertrochanteric or neck fracture)Ambulatory patients before fracture (with or without aid assistance), according to their New Mobility Score (NMS)Absence of any severe neuropsychiatric disorder (e.g., severe psychiatric disorder or dementia) to the extent that the researcher deems the patient incompetent or likely unable to remain compliant with the follow-up. All patients will be tested with Addenbrooke’s Cognitive Test III (ACE-III) to exclude those with moderate or severe dementia (<61 in total score) [27]Agreed to participate and signed the informed consent form

#### 2.4.2. Exclusion Criteria

Pathological fractures (on the ground of malignant musculoskeletal tumor)Unable to implement the exercise program due to underlying pathology or disability in the upper extremitiesMore than one fractureUnderlying avascular necrosisSevere and unstable cardiovascular disease (e.g., congenital heart disease, severe uncontrolled hypertension (systolic blood pressure ≥180 mmHg or diastolic blood pressure ≥120 mmHg) or unstable angina)Neurological or other conditions that significantly impair function and independence (e.g., stages 3–5 Parkinson’s disease based on the Hoehn and Yahr scale, advanced multiple sclerosis or severe osteoarthritis of degenerative or rheumatic etiology)Severe metabolic bone disease (e.g., Paget’s disease, renal bone disease or osteomalacia), excluding osteoporosisAny aggressive end-of-life disease (e.g., end-stage cancer with an estimated survival expectancy of less than six months)Unable to understand the informed consent form and protocol instructionsRefusal to participate in the research

#### 2.4.3. Withdrawal Criteria

Inability to participate in the inpatient program for more than two consecutive days (e.g., due to fever or early surgery within two days) or abstinence from more than 20% of the scheduled sessions in the postoperative programIntense subjective discomfort during exercise or development of other severe symptomatology or signs, as described in Section 2.4. Safety considerations and handling of adverse eventsPatient’s choice to drop out of the program

### 2.5. Procedures

Intervention group—Moderate-Intensity Interval Training (MIIT)

A supervised moderate-intensity interval training program will be applied using an upper-extremity cycle ergometer (Monark 881E Rehab Trainer, Monark, Vansbro, Sweden). The exercise program will be individualized by healthcare providers (physiotherapists and physicians) with more than five years of experience working with hip fracture patients. Additional external partners (physiotherapists) will be recruited for delivering the postoperative program. A commercially available athlete monitoring system app (Athlete Monitoring, Fitstats Inc., Moncton, NB, Canada) will be used for recording and documenting all training sessions’ features (intensity based on HR, duration, RPE, pain intensity and adverse events). The adherence rate will be calculated as the proportion of planned exercise sessions taken. Heart rate (HR) will be constantly recorded during each exercise session using an optical HR monitor (Polar OH1+, Polar Electro, Kempele, Finland). Oxygen saturation and blood pressure will also be assessed before and regularly during exercise. Pain intensity will be measured using a 20-point rating of perceived exertion scale (RPE) and will be considered as a limiting factor in maintaining sufficient pace and intensity in the exercise program.

The preoperative program will be performed in-hospital at the “Attikon” University Hospital. It will consist of 120 s of moderate-intensity exercise using an upper-extremity cycle ergometer and 120 s of passive rest for four cycles. The basic program will last about 16 min, not including the warm-up and cool-down. An additional 5 min in total is calculated for warm-up and recovery. The program will run twice a day. Exercise will be set at the half-seat position. The upper-extremity cycle ergometer will be set up on a rolling table, which will be placed over the bed in a comfortable position for the patient that allows a complete stroke without discomfort and prevents motion above the shoulders’ height. The adaptation and evaluation of the exercise intensity will be performed by modifying the external load (watts) based on the targeted internal load (moderate intensity: 64–76% predicted HRmax, 12–13 RPE). The revolutions per minute (rpm) will be set at 50 ± 10. Predicted HRmax will be estimated from the following formulae [34]:211—(0.64 × age in years) or
164—(0.7 × age in years), if taking beta-blocker

After each session, the lactate concentration of the capillary blood will be collected via a hand-held lactate analyzer (Lactate Scout 4, EKF Diagnostics, Leipzig, Germany), and the aim will be to maintain the capillary blood lactate concentration up to 2.0 mmol/L (±0.5).

The postoperative program will be performed initially in the hospital and at home after discharge. It will consist of 120 s of moderate-intensity exercise and 120 s of passive rest for eight cycles. The basic program will last about 30 min, not including warm-up and cool-down. An additional 5 min in total is calculated for warm-up and recovery. The program will be performed three times a week for eight weeks. The adaptation of the exercise intensity will be regulated by modifying the external load (watts) to keep the internal load of moderate intensity (64–76% HRmax, 12–13 in RPE). For the postoperative program, individuals will be allowed to follow the standard care, such as physical therapy sessions added to aerobic exercise. Standard care refers to postoperative rehabilitation. Such intervention will not be a part of our study’s protocol, and we will not intervene in patients’ decision to participate in physiotherapy sessions. These sessions will not be delivered by the research team, but it will be mentioned if they were conducted.

#### Attention Control Group

The control group will participate in a program of memory and attention activities of very low intensity with upper extremity movements using an interactive platform (Kinesthetic Multi-Sensory Games, Kinems, Greece). To maintain equal attention time with the intervention group, the program will last 20 min. There will be a simultaneous recording of the HR, while the RPE scale will be evaluated at the end of each session. For the postoperative program, no intervention will be performed beyond the standard care, such as physical therapy sessions without aerobic exercise.

### 2.6. Safety Considerations and Handling of Adverse Events

The inpatient and home exercise programs will be implemented and supervised by health professionals certified in first aid. Any intervention out of the hospital will begin after a cardiac clearance of the patient. For participating in the pre- and postoperative exercise program, approval by the attending orthopedic physician will be a prerequisite. During the initial assessment for planning and for any individualized modification to the planned exercise program to minimize potential adverse effects, the therapist will consider the following: current medical condition; medical history; comorbidities (e.g., diabetes and hypertension); medication (including dosage and time of administration); relevant clinical data (e.g., resting blood pressure and HR and oxygen saturation); instructions and cautions from other healthcare professionals; past and recent physical activity level time before the fracture and other factors that may affect exercise prescription (e.g., another injury).

#### 2.6.1. Absolute Contraindications for Exercise [35,36,37]

Progressive worsening of exercise tolerance or dyspnea at rest over previous 3–5 daysPresence of critical narrowing of the left coronary artery (obstructive left main coronary disease)Unstable anginaUncontrolled cardiac arrhythmiaAcute endocarditis, myocarditis or pericarditisModerate to severe aortic stenosisDecompensated heart failureAcute pulmonary embolism or deep-vein thromboembolismAortic dissectionHigh-grade atrioventricular blockHypertrophic obstructive cardiomyopathyRecent cerebrovascular accident (CVA) or transient ischemic attackUncontrolled diabetesRetinopathySevere autonomic or peripheral neuropathyAcute systemic illness or feverAcute or chronic renal failurePulmonary fibrosis or interstitial lung diseaseRecent myocardial infarction (<4 weeks), coronary artery bypass grafting (<4 weeks) or percutaneous coronary intervention

#### 2.6.2. Indications for Temporary Cessation

Symptoms such as angina, shortness of breath, dizziness (lightheadedness), confusion or signs of poor blood circulationOxygen saturation < 88%Increase in blood pressure > 220/105 mmHgDrop in systolic blood pressure > 10 mmHgHR decrease with higher load or development of any atypical arrhythmia

The therapist will be alert for any new symptoms or adverse events during exercise to discontinue the program and recommend referral to the physician and further investigation. Every adverse event will be recorded on the Athlete Monitoring app.

### 2.7. Outcome Measures

For all scales and questionnaires, permission to use has been granted after personal communication with the creators or with the team that has carried out the validation in the Greek population. All outcome measure measurements will be performed by an examiner blinded to the allocation group of the assessed (single-blinded). The demographic characteristics, fracture type, surgical technique and pre-fracture level of independence will be collected. All measurements will be performed in a lab setting.

The pre-fracture level of independence will be evaluated using the New Mobility Score (NMS). The NMS consists of three questions assessing the ability to walk in daily activities. The NMS is completed by the patient or by the caregiver in cases of cognitive impairment. The patient is asked to answer for pre-fracture walking ability the following: (1) able to move around the house (indoor walking), (2) able to go outside the house (outdoor walking) and (3) able to go shopping (walking while shopping). Each question can be scored from 0 to 3 points based on the patient’s self-reported ability, where 0 points correspond to an inability to perform, 1 point to assistance from another person, 2 points to use of an aid and 3 points to independent performance without assistance. The total score ranges from 0 to 9 points, with a score of 6 points or less indicating low functional capacity, while more than 6 points reflects a high level of pre-fracture function in patients with hip fractures. A score equal to or greater than 7 points justifies the rapid release of the patient from the hospital without the need for further in-hospital rehabilitation after surgery [38,39].

Table 1 represents the scheduled timeline of all the outcome measures.

#### 2.7.1. Basic Mobility

Basic mobility will be evaluated with the Cumulated Ambulation Score (CAS) and Timed-Up and Go (TUG). In orthopedic wards, the CAS is a reliable and valid measure for measuring patients’ mobility and is strongly recommended as an evaluation test by the Clinical Practice Guidelines of the American Physical Therapy Association (APTA) [1]. The CAS is an ordinal 3-point scale assessing a patient’s ability to get in and out of bed, sit to stand from a chair and walk inside a room. The scale for each test is scored from 0 to 2, with 0 meaning “unable to perform”, 1 “performing with assistance” and 2 “performing unassisted-independently”. The total 3-day CAS has a stronger predictive value than the 1-day CAS [40]; it is based on the sum score of the first three postoperative days, with a minimum value of 0 and a maximum value of 9; a higher score indicates a better prognosis for the total 3-day CAS [41,42]. An assessment will be performed on postoperative days 1, 2 and 3, as recommended by the literature, and at four weeks postoperatively.

Mobility will also be assessed with the TUG test. This test was designed to assess mobility in the elderly. The TUG test is a sensitive measure for predicting fall risks after hip fracture surgery [43]. It is a simple, easy-to-use functional performance test. The time required for the subject to rise from the chair, walk a distance of three meters, turn, walk back to the chair and sit is recorded. A chair (with back and supports), a tape measure, a tape and a clock or stopwatch are required. The assessment will be performed at 4, 8, 26 and 52 weeks postoperatively.

#### 2.7.2. Aerobic Capacity

The distance covered with the 6-minute walking test (6MWT) will be evaluated to assess their aerobic capacity. The 6MWT is an easy-to-use and reliable functional capacity assessment test. It is widely used and considered ideal for patients with respiratory, cardiovascular and other diseases characterized by reduced physical fitness [44,45]. It is simple to perform, easily understood by patients and reflects patients’ ability to cope with their daily activities, which mostly require submaximal work. The distance covered during the 6-minute walking test has been found to be positively correlated with the VO_2_ peak [46].

Patients are asked to walk as far as possible for 6 min. The patient will be allowed to use an external aid. There is strong evidence for the reliability and validity of the test in patients after hip fractures. Overgaard et al. [47] reported minimum detectable changes (MDC) without aid at 59.4 m and with aid at 49.8 m. The test will be performed along a 25 m aisle, and the HR will be simultaneously recorded using a HR monitor chest strap (Polar H10, Polar Electro, Kempele, Finland). The assessment will be performed at 4, 8, 26 and 52 weeks postoperatively.

#### 2.7.3. Lower Extremity Function

The modified Harris Hip Score (mHHS-Gr) and Lower Extremity Functional Scale (LEFS-Gr) will be used to determine lower extremity function. The mHHS [48] and LEFS [49,50] are valid and accurate self-reported measures of hip pain, function and lower extremity function. The mHHS has been used in femoral neck fractures with high discriminative capability between groups with and without complications at four months follow-up (95% CI Area Under the Curve: 0.63–0.89) and with good responsiveness from 4 to 12 months follow-up [51]. On the mHHS, one question assesses pain (score from 0–44) and seven questions functioning (score from 0–47). The total score will be multiplied by 1.1, resulting in a range from 0 to 100, with a score of 100 representing the worst possible hip pain and function. The LEFS comprises 20 multiple-choice questions scored on a 5-point scale (0–4: 0 = very high degree of difficulty to 4 = no difficulty, with a maximum score of 80). Assessment of the mHHS and LEFS will be performed on day 1 of admission to assess the pre-fracture level of independence (retrospective narrative) and at 4, 8, 26 and 52 weeks postoperatively.

#### 2.7.4. Blood Lactate Concentration

A hand-held lactate measuring analyzer (Lactate Scout+, EKF Diagnostics, Leipzig, Germany) with device-compatible test strips (sensors) will be used to obtain and analyze the lactate concentration of the capillary blood. Blood will be collected from the nail phalanx of the index hand with a puncture pen using sterile disposable lancets. Before each puncture, the nail phalanx will be antiseptic with gauze soaked in an alcoholic solution ≥70%. The measured range is from 0.5 to 25.0 mmol/L. The measurement accuracy is approximately ±3% when the hematocrit (Hct) ranges from 35 to 50% and ±4% when Hct > 50% [52,53]. The capillary blood lactate will be taken before and after each exercise session (only intervention group) to indicate the exercise intensity. The assessment will be performed on the day of admission, intraoperatively (at the beginning, 30 min later and at the end of the surgery) and 12 h postoperatively.

#### 2.7.5. Nutritional Status

To assess the patient’s nutritional status, the levels of albumin and prealbumin in the blood will be assessed. A blood sample will be collected and analyzed in the laboratory on the day of admission; one day before surgery; on the third postoperative day and at 1, 4, 8, 26 and 52 weeks postoperatively.

#### 2.7.6. Blood Sample Collection and Hemostatic Profile

The following factors will be evaluated: platelet functionality with a PFA-200 analyzer, hemostatic profile using viscoelastic techniques (ROTEM), tissue plasminogen activator (tPA), plasminogen activator inhibitor- 1 (PAI-1), the thrombin–antithrombin complex (TAT), the endogenous thrombin potential (ETP), fibrinogen, D-dimers, thrombomodulin and von Willebrand factors (vWF:Antigen and vWF:Activity plasma levels) as markers of endothelial function.

The standard range for fibrinogen is between 200 and 400 mg/dL (2.0 and 4.0 g/L). Plasma fibrinogen concentrations will be determined by modifying the Clauss technique with the Fibrinogen Multifibren U reagent (Siemens Healthcare Diagnostics, Marburg, Germany). Increased fibrinogen levels correlate substantially with inflammatory indicators, illness severity and thrombotic propensity.

The reference concentration of a D-dimer is <250 ng/mL. The particle-enhanced immunoturbidimetric INNOVANCE D-dimer assay (Siemens Healthcare Diagnostics, Marburg, Germany) will be used to evaluate the D-dimers. In addition to the value of normal D-dimer measurements in allowing the exclusion of venous thromboembolism (VTE) or pulmonary embolism (PE), the D-dimer test helps clinicians stratify patients with the risk for deep venous thrombosis (DVT), because high levels of D-dimer in the blood are associated with a major clot. Hence, elevated D-dimer levels indicate a hemostatic imbalance.

Normal thrombomodulin (TM) levels fluctuate between 3.1 ± 1.3 ng/mL and range from 3 to 300 ng/mL. For the measurements, a TM sandwich ELISA double antibody kit will be used. By controlling the coagulation system, TM, which is mainly expressed on the endothelium, plays a vital role in maintaining the vascular homeostasis. During the hypercoagulable condition after endothelial damage, TM is released into the intravascular space, suggesting endothelial dysfunction-related disruptions of the hemostatic balance. In addition, TM decreases blood coagulation by converting thrombin from a procoagulant to an anticoagulant enzyme, and it has the capacity to control intravascular damage via its pleiotropic actions.

vWF:Antigen (Ag) and vWF:Activity (Ac) are used as measures of platelet and endothelial dysfunction impacting the hemostatic equilibrium. The plasma levels of vWF:Ag and vWF:Ac will be measured using an automated latex enhanced immunoassay (HaemosIL^TM^ assay, Instrumentation Laboratory Corporation, Lexington, KY, USA) using IL (Instrumentation Laboratory) coagulation equipment (ACL TOP). The vWF:Ac will be evaluated by measuring the differences in turbidity caused by the latex reagent’s agglutination. A monoclonal anti-VWF antibody immobilized on the latex reagent interacts with the vWF in the blood sample. The agglutination is proportional to the vWF:Ac and is determined by the decreased light transmission resulting from the aggregates. The results will be presented as a percentage of the normality.

For the ROTEM analysis, within 90 min after blood collection, a citrated tube will be immediately filled with blood and will be analyzed in a ROTEM analyzer. Blood held at room temperature for up to 120 min after collection does not impact the ROTEM findings. Blood samples will be examined using a ROTEM analyzer (ROTEM delta, Tem Innovations GmbH). The ROTEM analysis includes preoperative EXTEM and INTEM testing and postoperative INTEM assays. The following EXTEM and INTEM parameters will recorded: clotting time (CT, seconds), the time from the beginning of measurement until the formation of a clot 2 mm in amplitude; clot formation time (CFT, seconds), the time from CT (amplitude of 2 mm) until a clot firmness of 20 mm was achieved; amplitude recorded at 10 min (A10, mm); an angle (a°), the angle between the central line and the tangent of the TEM tracing at the amplitude point of 2 mm describing the kinetics of clot formation; maximum clot firmness (MCF, mm), the final strength of the clot; and the lysis index at 60 min (LI60, %), which is the percentage of remaining clot stability in relation to the MCF following the 60-min observation period after CT, which indicates the speed of fibrinolysis [10].

Blood samples will be collected on the day of admission; one day before surgery; on the third postoperative day and at 1, 4, 8, 26 and 52 weeks postoperatively.

#### 2.7.7. Mortality and Incidence of Thrombotic/Thromboembolic Events

All thrombotic (e.g., myocardial infarction and stroke) and thromboembolic events (e.g., pulmonary embolism and deep vein thrombosis) will be recorded throughout the study. All-cause mortality will also be documented. Follow-up will take place up to 1 year after surgery.

#### 2.7.8. Perioperative Bleeding and Transfusion Requirements

Perioperative bleeding and transfusion requirements (liters) until discharge will be recorded.

#### 2.7.9. Readmission Rate for Any Reason

All-cause readmission within 30 days after surgery will be recorded and at all other assessment time points (8, 26 and 52 weeks postoperatively).

### 2.8. Statistical Analysis

#### 2.8.1. Sample Size Calculation

Power analysis for the sample size selection to highlight the preoperative program on the change in 3-day CAS was performed via the web application http://sampsize.sourceforge.net/ (accessed on 1 June 2021) for a significance level of α = 0.05 and power = 90%. The assumptions were made based on the study by Jérôme et al. [54], where m1 = 11.2 with a standard deviation = 3.1 and m2 = 13.6 with a standard deviation of 3.1. The minimal clinically important difference (MCID) between groups for the 3-day CAS was set at 2.4 units, given that the MCID for the 1-day CAS was found to be 0.8 units [39]. The ratio between groups will be 1:1. Due to factors such as in-hospital mortality, the drop-out rate is 10%. Considering the above assumptions, the minimum required sample to highlight the preoperative effectiveness of the exercise in the immediate postoperative basic mobility assessed with the 3-day CAS is set at 80 subjects (intervention group n1 = 40, control group n2 = 40).

The power analysis for sample size selection to demonstrate postoperative program change in the 6-minute walking test after program completion at eight weeks postoperatively was also performed via the web application: http://sampsize.sourceforge.net/ (accessed on 1 June 2021), with a significance level α = 0.05 and power = 90%. Based on previous studies [47,55,56], the 6MWT after a hip fracture at various time points (6 months, 20 days and one month postoperatively, respectively) varies approximately at 250 m of travel distance with a standard deviation of between 70 and 80 m. From the above-mentioned results, our assumptions for the power analysis, also assuming a clinically significant difference between the groups at 50 m, are m1 = 250 with a standard deviation = 75 and m2 = 300 with a standard deviation of 75. Considering the ratio between groups will be 1:1, the sample size is 96 patients (48 in each group).

However, based on the study by Karagiannis et al. [5] on a Greek population, mortality at six months after a fracture is 10.5%. However, both because this particular study reports a lower incidence of mortality at 6 and 12 months postoperatively compared to studies in other countries [57], and because of other factors that will likely contribute to the drop-out rate, such as comorbidities that will lead to the mandatory interruption of the program or the patient’s desire to interrupt the program for personal reasons, an increase in the sample size by 20% is deemed necessary, with the result that the sample size is set at 116 people (intervention group n1 = 58, control group n2 = 58).

Considering the above, in order to ensure the necessary power to investigate both primary purposes, the sample size is set at 116 subjects.

#### 2.8.2. Data Analysis

Descriptive statistics will be used to summarize the baseline characteristics of the patients participating in each group. Continuous variables will be presented with the mean and standard deviation or median and interquartile ranges (IQR) if they do not follow a normal distribution. Categorical variables will be presented with absolute and relative frequencies (n, %). The Kolmogorov–Smirnov test will be performed to check for normality. Based on the normal or non-normal distribution of the sample, a *t*-test or Mann–Whitney test will be performed to compare the means or medians of the two groups, respectively. The chi-square test will be used to identify differences in the categorical variables between the two groups. A subgroup analysis will be performed a posteriori for the different surgical techniques used for all variables.

To detect confounding factors that may affect the functional tests (CAS, 6WMT and TUG), multiple linear regression will be performed after first performing the relevant check on whether the conditions are met to consider the linear regression results reliable. If the necessary conditions are not met, the necessary transformations of the dependent variables will be carried out.

To test the change of the parameters to be measured at multiple time points (e.g., albumin, functional score, etc.) and to compare the change between the intervention group and the control group, we will perform an analysis of variance for repeated measurements (repeated measures ANOVA).

Additionally, the time to occurrence of an event (e.g., death, readmission to hospital, thrombosis, etc.) will be presented with a Kaplan–Meier survival curve. To highlight the factors related to the time of occurrence of an event, semiparametric (Cox proportional hazard models) or parametric survival analysis models (e.g., Weibull, exponential, etc.) will be performed, as will result from the analysis that will be done.

## 3. Discussion

This study has two primary purposes. First, to study the effect of a preoperative aerobic exercise program with a portable upper extremity cycle ergometer on immediate postoperative mobility. Second, to investigate the additional effect of an 8-week postoperative aerobic exercise program with a portable upper extremity cycle ergometer on the walking distance at 8 weeks after surgery. This study also has a number of secondary objectives. Regarding the preoperative exercise program evaluation of the effect of the preoperative program on surgical stress (via blood lactate levels), perioperative bleeding and transfusion requirements, hemostasis factors will be evaluated, in addition to, an evaluation of the effect of the postoperative program in the short and long term: lower extremity functionality, aerobic capacity, nutritional status, hemostasis factors, mortality, rehospitalization and complications related to surgery and hospitalization (thrombotic and thromboembolic events).

Undoubtedly after a hip surgery, postoperative rehabilitation is a well-established evidence-based practice [1,58,59]. The first days after the surgery seem to be the most critical for the patient’s subsequent independence. In a recent retrospective study, early mobilization (walking >1.5 m within 72 h postoperatively) was noted to be associated with fewer postoperative complications (myocardial infarction, pneumonia, admission to the intensive care unit or nursing home) and mortality within three months [60]. This has also been indicated by the high predictive value of the Cumulated Ambulation Score (CAS) [41]. Traditionally, the rehabilitation of hip fractures is based mainly on hip muscles strengthening [61,62] and improving mobility through gait retraining and functional tasks [1,63]. However, previous studies [17,64] have established the positive effect of aerobic exercise with a nonportable arm crank ergometer on outcomes such as the maximal oxygen uptake (VO_2_max), mobility and independence (Timed Up and Go test and 6-min walking distance), functionality and quality of life of patients after a hip fracture. Similar improvements in the outcome measures have been demonstrated after total hip arthroplasty, with favorable results relative to the control group being maintained up to one year after surgery [65]. Although intensity is an important factor for exercise effectiveness, and most published exercise protocols refer to moderate-intensity continuous training for elderly individuals, moderate-intensity interval training seems sufficient for functional and metabolic improvements in older adults [66,67]. Our pilot trial has revealed that interval training may be more appropriate and feasible for this population, considering the advanced age, comorbidities and lack of exercise experience these patients have.

There is less evidence about the value of prehabilitation, which is, however, suggested as appropriate, or even necessary, both for the prevention of postoperative complications and the best outcome of surgery [68,69], as well as for the favorable cost-effectiveness it offers [70]. Based on the existing literature on both the benefits of aerobic exercise on cardiorespiratory health and functionality in patients with hip fractures [17,64], the positive regulation it brings about in biochemical markers such as hypercoagulability [22,23] and albumin [71,72], we hypothesize that a pre- and postoperative exercise program can provide a comprehensive, multidimensional and multisystemic benefit to patients after a hip fracture. Delayed surgery, beyond 48 h of admission, is a worldwide issue [73]. Although, in regional hospitals, the length of waiting time might be considered during early surgery on average [74], in major hospitals, almost one-third are delayed surgeries in Greece [75]. Therefore, considering the Greek reality where, in hospital clinics, patients’ surgeries are delayed due to a disproportion in the number of cases and staff, resulting in a possible impact on patient outcomes [76], there is an immediate need to investigate the dynamics of such a conservative intervention in biochemical markers related to morbidity and mortality.

Although postoperative (post-hospitalization) moderate-intensity aerobic exercise programs have been used in hip fracture patients and tested for safety and efficacy [17,64], these patients typically fail to incorporate the minimum weekly physical activity guidelines into their rehabilitation program, as recommended by the World Health Organization, both during their hospitalization and during their later life in the community [77,78]. To date, aerobic exercise rehabilitation programs for these patients have relied on nonportable upper extremity cycle ergometers. This implementation is a critical limiting factor in the applicability and feasibility of such programs in the real world, given that, for older adults, the inability to move outside the home is the second-most important factor for not participating in an exercise program, with the most important being a lack of interest in exercise itself [79,80]. In a recent retrospective observational study of 681 hip fracture patients, preoperative implementation of a physical therapy program of lower extremity strengthening and range of motion (in the unaffected extremity) showed a statistically significant improvement in the level of independence at discharge in comparison with the control group [81].

### Expected Outcome

For the preoperative exercise program, we hypothesize that it will reduce surgical stress through the body’s better response to lactate accumulation and acute physiologic effects on the patient’s hematopoietic activity. Exercise will reduce platelet hyperactivity, increase the activity of the fibrinolytic system and decrease the procoagulant factors. In this more favorable environment, the immediate postoperative restoration of basic mobility will be able to be accelerated. Regarding the postoperative exercise program, we hypothesize that the aerobic exercise program will improve lower extremity independence and functionality by maintaining or increasing the cardiopulmonary capacity, while improving the nutritional factors through post-exercise protein synthesis will contribute to this.

## Figures and Tables

**Figure 1 biomedicines-11-01263-f001:**
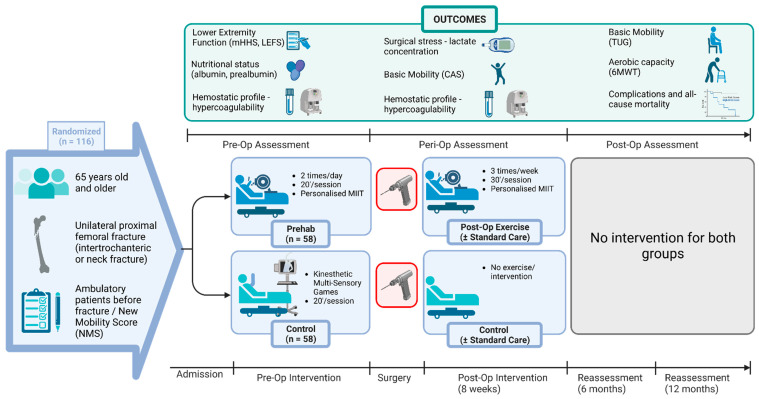
Graphical illustration of the study design. In the frame “outcomes”, only the first integration of the assessments and not the follow-up assessment can be seen. For example, mHHS will be assessed not only at admission (first integration as shown in the box) but also at 4, 8, 26 and 52 weeks (follow-up assessment). Accordingly, the evaluation of the basic mobility through the CAS scale will be evaluated on the first; second and third postoperative day and at 4, 8, 26 and 52 weeks postoperatively. Blood samples will be collected on the day of admission; one day before surgery; on the third postoperative day and at 1, 4, 8, 26 and 52 weeks postoperatively. Analysis of the coagulation biomarkers will be done by ROTEM. After the 8 weeks of postoperative intervention, there will be no intervention for any of the two groups. The time from admission to surgery will vary. Created with BioRender.com (accessed on 16 April 2023). Abbreviations: MIIT = Moderate Intensity Interval Training; CAS = Cumulative Ambulation Score; 6MWT = six-minute walk test; TUG = Timed-Up and Go; mHHS = modified Harris Hip Score; LEFS = Lower Extremity Functional Scale; NMS = New Mobility Score; La = Lactate concentration of capillary blood; ROTEM = rotational thromboelastometry; Hemostatic profile = tissue Plasminogen Activator (tPA); Plasminogen Activator Inhibitor-1 (PAI-1); Endogenous Thrombin Potential (ETP); Fibrinogen; D-dimers; von Willebrand factor antigen; von Willebrand factor activity; Thrombin–Antithrombin Complex (TAT); Thrombomodulin; platelet function—PFA-100; hemostatic profile using viscoelastic techniques—ROTEM; pre-op = preoperative; peri-op = perioperative; post-op = postoperative.

**Table 1 biomedicines-11-01263-t001:** Time points of the outcome measurements. In bold are the primary outcome measures for which the sample size was calculated for the effects of the preoperative and the postoperative exercise program.

Outcome Measures	1st Day of Admission	One Day before Surgery	During Surgery	Post-Op—to Discharge	4th Week Post-Op	8th Week Post-Op	26th Week Post-Op	52nd Week Post-Op
**CAS**				**1st, 2nd, 3rd post-op**	**✓**			
**6MWT**					**✓**	**✓**	**✓**	**✓**
TUG					✓	✓	✓	✓
mHHS	✓				✓	✓	✓	✓
LEFS	✓				✓	✓	✓	✓
NMS	✓							
La	✓		✓	12 h post-op				
Nutrional status	✓			3rd post-op	1st and 4th wk	✓	✓	✓
Hemostatic factors	✓			3rd post-op	1st and 4th wk	✓	✓	✓
Bleeding and transfusion needs			✓	✓				
Readmission					✓	✓	✓	✓
All-cause mortality and complications		✓			✓	✓	✓	✓

CAS = Cumulative Ambulation Score; 6MWT = six-minute walk test; TUG = Timed-Up and Go; mHHS = modified Harris Hip Score; LEFS = Lower Extremity Functional Scale; NMS = New Mobility Score; La = Lactate concentration of capillary blood; Nutritional status = Albumin and prealbumin; Hemostatic factors = tissue Plasminogen Activator (tPA); Plasminogen Activator Inhibitor-1 (PAI-1); Endogenous Thrombin Potential (ETP); Fibrinogen; D-dimers; von Willebrand factor antigen; von Willebrand factor activity; Thrombin–Antithrombin Complex (TAT); Thrombomodulin; platelet function—PFA-100; hemostatic profile using viscoelastic techniques—ROTEM; post-op = postoperative.

## Data Availability

The data of this study will be available from the corresponding author upon reasonable request. The data are not publicly available due to containing information that could compromise the privacy of the research participants.

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
