# Peer review of "Pre- and Postoperative Exercise Effectiveness in Mobility, Hemostatic Balance, and Prognostic Biomarkers in Hip Fracture Patients: A Study Protocol for a Randomized Controlled Trial"

_biomedicines, 2023, doi:10.3390/biomedicines11051263_

Round 1
Reviewer 1 Report
BIOMEDICINES-2316332 presents a protocol for an RTC. Given that this is a developed protocol, my comments are focused on presentation of work. I hope the authors consider my feedback.
· Line 65: Avoid the use of “cause” type language here and elsewhere.
· It is the opinion of this reviewer that the information presented in Figure 1 should align with the Introduction better. For example, many blood biomarkers are listed in the introduction are not directly listed on Figure 1 (again in my opinion). This comment may have applicability to parts of the discussion. Consider revision.
· Not sure if some of the details in lines 98-106 are needed.
· Line 116: Please delete, “(G.K.)”.
· Section 2.2: What about plans for mitigating attrition?
Author Response
Thanks for your suggestions!
General comments: BIOMEDICINES-2316332 presents a protocol for an RTC. Given that this is a developed protocol, my comments are focused on presentation of work. I hope the authors consider my feedback.
Response to general comments: Thank you for your valuable suggestions.
Point 1: Line 65: Avoid the use of “cause” type language here and elsewhere.
Response 1: We changed “cause” to “are associated with.“ (page 2, line 83)
Point 2: It is the opinion of this reviewer that the information presented in Figure 1 should align with the Introduction better. For example, many blood biomarkers are listed in the introduction are not directly listed on Figure 1 (again in my opinion). This comment may have applicability to parts of the discussion. Consider revision.
Response 2: We recreated Figure 1 using Biorender following your suggestions. We added blood biomarkers as a caption, considering they are too many. We also added some more information in the Introduction. (pages 2-3, lines 98-101)
Point 3: Not sure if some of the details in lines 98-106 are needed.
Response 3: We consider the following information necessary to maintain: “The study protocol has been prospectively registered in the open database for study protocols ClinialTrials.gov (registration number: NCT05389800) [28]. The study received approval number 402/26-7-2021 from the Institutional Review Board of the “Attikon” University Hospital (1st Department of Orhtopaedic Surgery, National and Kapodistrian University of Athens) September 7, 2021.” (page 3, lines 119-127)
Point 4: Line 116: Please delete, “(G.K.)”.
Response 4: Done. (page 3, line 137)
Point 5: Section 2.2: What about plans for mitigating attrition?
Response 5: We added some information as suggested. (page 7, lines 173-176)
Reviewer 2 Report
Journal MDPI Biomedicines, study protocol
Title of paper Pre- and post-operative effects of upper extremity aerobic exercise in mobility, hemostatic balance, and mortality- and morbidity-related biomarkers in hip fracture patients: a study protocol for a randomized controlled trial
General comments
The theme of the study is topical and interesting.
The size of the possible groups for study (n=58 in each group) is sufficient for valid conclusions, the results will be obtained with objective methods used in physiotherapy practice.
The study design is presented in one figure and one table.
The inclusion, exclusion and withdrawal criteria of participants, as well as contraindications for exercise performance and indication for temporary cessation have been adequately described.
The intervention methods concerning moderate intensity interval training (MIIT) for pre-surgery and post-operative 8-week period have been adequately described.
The measures for assessing mobility (scores and TUG test), aerobic capacity (6MWT) and lower extremity muscle function (scores) will be applied.
The set of potential blood markers and principles for their analysis have been added.
Aspects that need further argumentation or explanation:
1) Pre-operative exercise program duration – is it really possible to have 4 MIIT exercise sessions from the day of being admitted to hospital to the day of surgery.
2) The number of pre-operative exercise sessions is not sufficient for demonstrating their effect.
3) Pain as a limiting factor in MIIT performance before surgery has to be considered.
Specific comments:
Title: It is recommended to shorten the title.
Abstract: Please add information about the number of patients in the control group and about MIIT.
Introduction
It is recommended to begin with hip fracture occurrence survey in different age decades.
Please add information about the application of MIIT exercise, and provide more focus on the rehabilitation after hip fracture.
L58 The correct term is prealbumin (not proalbumin).
Participants
It is recommended to include this chapter in the article, providing information on the number of patients in both groups, their gender and age range.
Inclusion criteria:
L134-135 Outpatient department patients – how many patients and in which way it is planned to study?
L 139-143 Cognitive function level – it is recommended to add here ( from the exlusion criteria of patients).
What about avascular necrosis and pulmonal diseases in this respect?
L253 What are the recommended lowest systolic and diastolic blood pressure indices?
Fig 1. Study design
For both groups should be included the stage-by-stage scheme of the study design, including pre-operative and post-operative intervention. Currently only partial survey of the study design has been presented.
Interventions
It is recommended to provide the survey of interventions in both groups and add this information on the study design figure as well.
- Standard physiotherapy. Will it be supervised? Is it a home exercise program or conducted in outpatient department? What is the duration of the program? Will patients keep the respective diary?
Is this program similar for both groups?
- MIIT exercises. Will they be supervised? Where are they performed - in lab conditions or at home?
Will the exercises be performed in the sitting or lying position and in which manner will the program be organised?
Outcome measurements
Where will the measurements be performed – in lab conditions?
Table 1. Time points of measurements
Lactate level during surgery – what is the reason for including this characteristic?
Readmission – can this be decided during surgery?
Bleeding and transfusion needs – when should attention be paid to these needs?
All-cause mortality and complications – why is this assessed one day before surgery, not on the 1st day of admission to hospital?
Statistics
Data analysis
It is recommended to add correlation analysis.
2.7.3. Final reports – this part should be shortened.
4. Patents – please add some comments or delete this part.
Author Response
We appreciate your thorough review of our article and the time you took for this. Please see our point-by-point response.
General comments: The theme of the study is topical and interesting. The size of the possible groups for study (n=58 in each group) is sufficient for valid conclusions, the results will be obtained with objective methods used in physiotherapy practice. The study design is presented in one figure and one table. The inclusion, exclusion and withdrawal criteria of participants, as well as contraindications for exercise performance and indication for temporary cessation have been adequately described. The intervention methods concerning moderate intensity interval training (MIIT) for pre-surgery and post-operative 8-week period have been adequately described. The measures for assessing mobility (scores and TUG test), aerobic capacity (6MWT) and lower extremity muscle function (scores) will be applied. The set of potential blood markers and principles for their analysis have been added.
Response to general comments: We appreciate your thorough review of our article and the time you took for this. Your comments were to the point for improving our manuscript. We revised our manuscript following your suggestions.
---
Aspects that need further argumentation or explanation:
Point 1: Pre-operative exercise program duration – is it really possible to have 4 MIIT exercise sessions from the day of being admitted to hospital to the day of surgery.
Response 1: Actually, in Greece, delayed surgery in patients with hip fractures is common. For example, in Thessaloniki (the second biggest city after Athens), delayed surgery was found (> 48 hours) to be as high as 67.6%, with an average of 6.06 days since admission.
Chatziravdeli, V., Vasiliadis, A. V., Vazakidis, P., Tsatlidou, M., Katsaras, G. N., & Beletsiotis, A. (2021). The Financial Burden of Delayed Hip Fracture Surgery: A Single-Center Experience. Cureus, 13(3), e13952.
Another study in a regional hospital (Larissa) indicates a similar percentage of delayed surgery (54.1%).
Dailiana, Z., Papakostidou, I., Varitimidis, S., Michalitsis, S., Veloni, A., & Malizos, K. (2013). Surgical treatment of hip fractures: factors influencing mortality. Hippokratia, 17(3), 252–257.
In Athens, where our study will be conducted, delayed surgery might be present at an even higher rate (unpublished-unofficial data).
Point 2: The number of pre-operative exercise sessions is not sufficient for demonstrating their effect.
Response 2: Indeed, the number of exercise sessions in terms of physiological adaptations might is not sufficient. Although existing data suggest that a training period of at least 3 to 4 weeks is required to induce a significant increase in cardiac mass, there is some evidence of significant neuromuscular
responses within just three days of specific exercise or cardiovascular adaptations within 10 days of exercise, enhancing the inotropic response to β-adrenergic stimulation, associated with increases in cardiac output and stroke volume during peak exercise.
• Coburn, J. W., Housh, T. J., Malek, M. H., Weir, J. P., Cramer, J. T., Beck, T. W., & Johnson, G. O. (2006). Neuromuscular responses to three days of velocity-specific isokinetic training. Journal of strength and conditioning research, 20(4), 892–898.
• Mier, C. M., Turner, M. J., Ehsani, A. A., & Spina, R. J. (1997). Cardiovascular adaptations to 10 days of cycle exercise. Journal of applied physiology (Bethesda, Md. : 1985), 83(6), 1900–1906.
However, there is compelling evidence that just 1 week of pre-operative exercise provides significant protection against postoperative pulmonary complications, might prevent post-operative functional decline, and may have beneficial effect on hemostatic balance.
• Assouline, B., Cools, E., Schorer, R., Kayser, B., Elia, N., & Licker, M. (2021). Preoperative Exercise Training to Prevent Postoperative Pulmonary Complications in Adults Undergoing Major Surgery. A Systematic Review and Meta-analysis with Trial Sequential Analysis. Annals of the American Thoracic Society, 18(4), 678–688.
• Sebio García, R., Yáñez-Brage, M. I., Giménez Moolhuyzen, E., Salorio Riobo, M., Lista Paz, A., & Borro Mate, J. M. (2017). Preoperative exercise training prevents functional decline after lung resection surgery: a randomized, single-blind controlled trial. Clinical rehabilitation, 31(8), 1057–1067.
• Skouras, A.Z.; Antonakis-Karamintzas, D.; Tsantes, A.G.; Triantafyllou, A.; Papagiannis, G.; Tsolakis, C.; Koulouvaris, P. The Acute and Chronic Effects of Resistance and Aerobic Exercise in Hemostatic Balance: A Brief Review. Sports 2023, 11, 74.
Point 3: Pain as a limiting factor in MIIT performance before surgery has to be considered.
Response 3: In section 2.4. we added that pain would be measured using a 20-point rating of perceived exertion scale (RPE) and would be considered as a limiting factor in maintaining sufficient pace and intensity in exercise program. [page 8, lines 235-237]
---
Specific comments:
Point 4: It is recommended to shorten the title.
Response 4: We shorted our title from 30 words to 24, as follows “Pre- and Post-Operative Exercise Effectiveness in Mobility, Hemostatic Balance, and Prognostic Biomarkers in Hip Fracture Patients: A Study Protocol for a Randomized Controlled Trial.” [page 1, lines 2-6]
Point 5: Please add information about the number of patients in the control group and about MIIT.
Response 5: It is not clear if you mean in abstract or in general.
We added in the abstract that
“A […] RCT is planned to be conducted with 58 patients each in the intervention and control groups.” [page 1]
The main text reports the number of patients in the control group in Study Design (section 2.1). [page 3, line 140]
About MIIT, in the abstract, we added that we will perform MIIT and the following two sentences:
“The work-to-recovery ratio will be 1-to-1, consisting of 120 seconds for each bout, and 4 and 8 rounds for pre- and post-operative programs, respectively. Preoperative program will be delivered twice a day.” [page 1]
In the main text (section 2.4), we have already fulfilled all checklist points as referred on Consensus on Exercise Reporting Template (CERT). [page 8, lines 222-268]
---
Introduction:
Point 6: It is recommended to begin with hip fracture occurrence survey in different age decades.
Response 6: We transferred and modified a relevant paragraph from the Discussion to the Introduction. We revised the whole first paragraph, adding hip fracture epidemiology data, focusing on Greece. [page 2, lines 52-66]
Point 7: Please add information about the application of MIIT exercise, and provide more focus on the rehabilitation after hip fracture.
Response 7: Regarding rehabilitation, we already had written some information in the section Discussion. We also added some more evidence about rehabilitation and MIIT in this section. If you consider it necessary to provide more information or to transfer this paragraph to the Introduction section, we are open to managing it. [page 17, lines 629-631 and 637-642]
Point 8: L58 The correct term is prealbumin (not proalbumin).
Response 8: We corrected it. [page 2, line 75]
---
Participants:
Point 9: It is recommended to include this chapter in the article, providing information on the number of patients in both groups, their gender and age range.
Response 9: Done (section 2.2). [page 3-4, lines 145-153]
---
Inclusion criteria:
Point 10: L134-135 Outpatient department patients – how many patients and in which way it is planned to study?
Response 10: It is not clear what you mean by “outpatient department patients.” All patients will be collected from the Emergency Department with acute hip fractures. New Mobility Score (NMS) will be used to determine the pre-operative independence status, and patients who were not in sufficient mobility and function before the fracture (non-ambulatory) will not be included in the study from the beginning.
Point 11: L 139-143 Cognitive function level – it is recommended to add here ( from the exlusion criteria of patients).
Response 11: Done. [page 7, lines 183-187]
Point 12: What about avascular necrosis and pulmonal diseases in this respect?
Response 12: Avascular necrosis has been added to the exclusion criteria [page 7, line 199]. We do not really have any pulmonary disease in our mind that we should exclude from our study.
Point 13: L253 What are the recommended lowest systolic and diastolic blood pressure indices?
Response 13: We do not have the lowest values because our exercise program will be delivered in a semi-seated position. Thus, there is no concern for arterial hypotension symptoms, such as dizziness or drop attack.
---
Fig 1. Study design
Point 14: For both groups should be included the stage-by-stage scheme of the study design, including pre-operative and post-operative intervention. Currently only partial survey of the study design has been presented. It is recommended to provide the survey of interventions in both groups and add this information on the study design figure as well.
Response 14: We recreated Figure 1 following your recommendations. [page 6]
---
Interventions
Point 15: Standard physiotherapy. Will it be supervised? Is it a home exercise program or conducted in outpatient department? What is the duration of the program? Will patients keep the respective diary? Is this program similar for both groups?
Response 15: “Standard care refers to post-operative rehabilitation. Such intervention will not be a part of our study’s protocol, and we will not intervene in patients’ decision to participate in physiotherapy sessions. These sessions will not be delivered by the research team, but it will be mentioned if they were conducted.” [page 8, lines 264-268]
Point 16: MIIT exercises. Will they be supervised? Where are they performed - in lab conditions or at home?
Response 16: All these details are already mentioned on the following points:
Page 8, Line 223: “A supervised moderate-intensity interval training program will be applied”.
Page 8, Lines 238-239: “The preoperative program will be performed in-hospital at the “Attikon” University Hospital.”
Page 8, Lines 256-257 : “The post-operative program will be performed initially in the hospital and at home after discharge.”
Point 17: Will the exercises be performed in the sitting or lying position and in which manner will the program be organised?
Response 17: Exercise will be set at half seat position. The upper-extremity cycle ergometer will be set up on a rolling table which will be placed over the bed in a comfortable position for the patient that allows a complete stroke without discomfort and prevents motion above the shoulders’ height. Revolution per minute (rpm) will be set at 50 ± 10. [page 8, lines 243-246 and 248-249]
---
Outcome measurements
Point 18: Where will the measurements be performed – in lab conditions?
Response 18: All measurements will be performed in lab setting. [page 10, line 328]
---
Table 1. Time points of measurements
Point 19: Lactate level during surgery – what is the reason for including this characteristic?
Response 19: By definition, surgical stress is the systemic response to surgical injury and is characterized by activation of the sympathetic nervous system, endocrine responses, as well as immunological and hematological changes.
“An increase in intraoperative lactate, independent of the level on induction, is a useful dynamic parameter to identify patients at risk of postoperative morbidity and mortality and might provide an early trigger for introducing measures to avoid poor outcomes.”
• Govender, P., Tosh, W., Burt, C., & Falter, F. (2020). Evaluation of Increase in Intraoperative Lactate Level as a Predictor of Outcome in Adults After Cardiac Surgery. Journal of cardiothoracic and vascular anesthesia, 34(4), 877–884.
• Duval, B., Besnard, T., Mion, S., Leuillet, S., Jecker, O., Labrousse, L., Rémy, A., Zaouter, C., & Ouattara, A. (2019). Intraoperative changes in blood lactate levels are associated with worse short-term outcomes after cardiac surgery with cardiopulmonary bypass. Perfusion, 34(8), 640–650.
Lactate level 12 hours post-operatively has been revealed to have the best power to discriminate between patients with and without postoperative complications after elective surgery and was associated with a longer hospital stay.
• Veličković, J., Palibrk, I., Miličić, B., Veličković, D., Jovanović, B., Rakić, G., Petrović, M., & Bumbaširević, V. (2019). The association of early postoperative lactate levels with morbidity after elective major abdominal surgery. Bosnian journal of basic medical sciences, 19(1), 72–80.
Point 20: Readmission – can this be decided during surgery?
Response 20: The tick symbol had been put in the timeline table by mistake. We removed it and added it to the previous row, “bleeding and transfusion needs.” [page 11]
Point 21: Bleeding and transfusion needs – when should attention be paid to these needs?
Response 21: Until hospital discharge, we will document any blood transfusion need. We had not put the tick symbol on the timeline table by mistake. [page 11]
Point 22: All-cause mortality and complications – why is this assessed one day before surgery, not on the 1st day of admission to hospital?
Response 22: General health information will be collected during the pre-operative check the day before surgery. Also, on the 1st day of admission, we cannot see how all-cause mortality or any complication (due to surgery, exercise, or any other reason) reporting will be applicable from the 1st day of admission.
---
Statistics
Data analysis
Point 23: It is recommended to add correlation analysis.
Response 23: To the best of our knowledge, correlation analysis is used to identify any association between continuous variables. Given that the objectives of our study are to identify any differences between the two groups, the correlation analysis may is not applicable. If you still think some correlation analysis is required, please provide us with more clarification.
---
Point 24: 2.7.3. Final reports – this part should be shortened.
Response 24: We shortened its length from 208 to 114 words. [page 16, lines 559-569]
Point 25: 4. Patents – please add some comments or delete this part.
Response 25: It is not clear to us why we need to delete this section. Author contributions, funding, IRB, informed consent and data availability statements, and conflicts of interest are presented here.
Reviewer 3 Report
The study protocol is well written. It has been registered on an open database. Furthermore, the illustration is clear. The framework is explicitly depicted. I would like to endorse its publication. I only have a minor suggestion. Can the authors add a small session of "expected outcome"? The added portion will make the whole article more complete.
Author Response
Thanks for your time and suggestions. Please see our response.
General comments: The study protocol is well written. It has been registered on an open database. Furthermore, the illustration is clear. The framework is explicitly depicted. I would like to endorse its publication.
Response to general comments: Thank you for your kind comments.
Point 1: I only have a minor suggestion. Can the authors add a small session of "expected outcome"? The added portion will make the whole article more complete.
Response 1: Yes, we created a new sub-section in the Discussion. There we transferred our whole hypotheses from the Discussion. We are open to revising properly if you consider that more information and hypotheses are needed. [page 18, lines 674-683]